# Enhancing Greenhouse Efficiency: Integrating IoT and Reinforcement Learning for Optimized Climate Control

**DOI:** 10.3390/s24248109

**Published:** 2024-12-19

**Authors:** Manuel Platero-Horcajadas, Sofia Pardo-Pina, José-María Cámara-Zapata, José-Antonio Brenes-Carranza, Francisco-Javier Ferrández-Pastor

**Affiliations:** 1Informática Industrial y Redes de Computadores (I2RC), University of Alicante, 03690 Alicante, Spain; 2Centro de Investigación e Innovación Agroalimentario y Agroambiental (CIAGRO), Miguel Hernandez University, 03312 Orihuela, Spain; spardo@umh.es (S.P.-P.); jm.camara@umh.es (J.-M.C.-Z.); 3Centro de Investigaciones en Tecnologías de la Información y Comunicación (CITIC), University of Costa Rica, San José 11501, Costa Rica; joseantonio.brenes@ucr.ac.cr

**Keywords:** smart agriculture, reinforcement learning, IoT, greenhouse energy management

## Abstract

Automated systems, regulated by algorithmic protocols and predefined set-points for feedback control, require the oversight and fine tuning of skilled technicians. This necessity is particularly pronounced in automated greenhouses, where optimal environmental conditions depend on the specialized knowledge of dedicated technicians, emphasizing the need for expert involvement during installation and maintenance. To address these challenges, this study proposes the integration of data acquisition technologies using Internet of Things (IoT) protocols and optimization services via reinforcement learning (RL) methodologies. The proposed model was tested in an industrial production greenhouse for the cultivation of industrial hemp, applying adapted strategies to the crop, and was guided by an agronomic technician knowledgeable about the plant. The expertise of this technician was crucial in transferring the RL model to a real-world automated greenhouse equipped with IoT technology. The study concludes that the integration of IoT and RL technologies is effective, validating the model’s ability to manage and optimize greenhouse operations efficiently and adapt to different types of crops. Moreover, this integration not only enhances operational efficiency but also reduces the need for constant human intervention, thereby minimizing labor costs and increasing scalability for larger agricultural enterprises. Furthermore, the RL-based control has demonstrated its ability to maintain selected temperatures and achieve energy savings compared to classical control methods.

## 1. Introduction

The integration of IoT and RL creates an intelligent and adaptive control system for greenhouses. IoT sensors provide a constant flow of accurate information, which RL algorithms then process to optimize growing conditions. Automated systems implement the decisions made by RL, and the system constantly improves on the basis of the results obtained. This synergy allows for more efficient resource management, increased productivity, and better adaptation to changing conditions, which is crucial in the current context of climate change and food security. By combining the data collection capabilities of IoT with the decision-making power of RL, greenhouse operators can create a self-improvement system that continuously adapts to changing environmental conditions and plant needs. This integration not only improves crop yield and quality but also contributes to sustainable agriculture practices by optimizing resource use. As climate change continues to pose challenges to traditional farming methods, the integrated IoT-RL approach in greenhouses offers a promising solution to ensure food security in an environmentally responsible way. This paper is structured as follows. Section 2 provides a comprehensive review of the literature, highlighting key contributions and future research directions in the use of reinforcement learning (RL) for energy management in greenhouses. Section 3 presents materials and methods. In Section 3.1, we present the proposed model, detailing the integration of IoT and RL technologies for optimized climate control in greenhouses. Section 3.2 discusses the methodology, including the phases of development of control strategies, IoT infrastructure design, dataset generation, and the application of digital models and RL algorithms. The IoT infrastructure is proposed in Section 3.3. In Section 3.4, we analyze the generation of dataset and the materials used. Section 3.5 proposes the RL algorithm based on the model. Section 3.6 implements and evaluates the RL agent, comparing its performance with traditional control methods. Finally, Section 4 and Section 5 conclude the article, summarizing the findings and discussing the implications for future research and practical applications in greenhouse management.

Automated greenhouse systems require the meticulous configuration of diverse parameters to guarantee that the actuators execute the designated functions accurately. This configuration entails the determination of appropriate set-points for various subsystems including climate, lighting, and irrigation control. Typically, these parameters and control rules are reactive. In recent years, numerous studies have advanced models based on set-point selection strategies employing various heuristic or artificial intelligence paradigms coupled with the implementation of predictive solutions. The application of reinforcement learning (RL) to greenhouse management has gained significant attention in recent years due to its potential to optimize energy usage and automate various control processes. This section provides an overview of the most relevant studies categorized by their approaches and contributions.

## 2. Literature Review

The use of reinforcement learning (RL) in the energy management of greenhouses and precision agriculture techniques has evolved significantly. This analysis is based on the references provided, which highlight key contributions in this field. Future research directions are also discussed. This paper reviews key contributions to the field, analyzes their contributions, and discusses future research directions.

Kiumarsi et al. [1] present a comprehensive survey and implementation guidelines for optimal and autonomous control using RL. They discuss applications in complex systems, emphasizing their potential to enhance energy efficiency and reduce operational costs in greenhouses. Perera et al. [2] review various applications of RL in energy management from generation to consumption. They highlight specific cases of intelligent greenhouses that use RL to optimize energy use. Wang et al. [3] present an RL controller based on recurring neural networks based on Long Short-Term Memory (LSTM) for microgrid management with potential applications in greenhouses.

Kazmi et al. [4] explore the use of deep RL for optimal control in hybrid energy systems of buildings, which is applicable to energy management in greenhouses. Ruelens et al. [5] examine the application of RL in electric water heaters with direct implications for energy management in greenhouses.

Zhang et al. [6] discuss the developments and future challenges in precision agriculture, including the integration of RL for better resource management in greenhouses. Liu et al. [6] provide a comprehensive review of RL applications in smart agriculture, highlighting various use cases and the potential benefits of RL in optimizing agricultural processes, including greenhouse energy management.

Mason et al. [7] review the applications of RL in smart grids, discussing how these methods can be applied to improve energy management and efficiency in interconnected systems such as greenhouses. Hosseinloo et al. [8] explore data-driven predictive control using RL for energy efficiency and comfort management in buildings with potential applications in greenhouses to optimize climate control systems. Sun et al. [9] present a study on multi-agent RL for the integrated energy management of interconnected microgrids, which is applicable to complex greenhouse energy systems. Alani et al. [10] discuss the opportunities and challenges of RL-based energy management for smart homes and buildings with insights applicable to greenhouse energy systems.

Fu et al. [11] survey the applications of RL in building energy management, providing information relevant to greenhouse energy systems. Mauree et al. [12] review data-driven and machine learning models for building energy performance prediction, fault detection, and optimization, all of which are applicable to greenhouse energy management. Kazmi et al. [13] discuss the application of multi-agent RL for building energy management with potential benefits for greenhouse systems. Yang et al. [14] explore energy optimization in smart home buildings using deep RL, providing information for greenhouse energy management.

Ruelens et al. [15] discuss learning sequential decision making for the optimal control of thermally activated resources, which is applicable to greenhouse energy systems. Vázquez-Canteli and Henze [16] discuss the integration of reinforcement learning with predictive control of the model for the response to demand in buildings, which can be applied to optimize energy management in greenhouses. Sutton and Barto [17] discuss the application of RL in real-world games, providing insights that can be adapted to complex energy management systems in greenhouses. Peters et al. [18] discuss a reinforcement learning approach to autonomous vehicles, which can provide insights for autonomous control in greenhouse energy systems. Kar et al. [19] present QD learning, which is a collaborative Q-learning approach that can be applied to cooperative energy management strategies in greenhouses.

Sierla et al. [20] review reinforcement learning applications in urban energy systems, which can provide valuable information for energy management in greenhouse environments. Vázquez-Canteli and Nagy [21] discuss reinforcement learning for demand response, which is highly relevant for dynamic energy management in greenhouses. Mauree et al. [12] review assessment methods for urban environments, providing methodologies that can be adapted to evaluate energy performance in greenhouses.

### 2.1. Future Research Directions

Despite significant advancements, several areas require further exploration to fully implement RL in greenhouse energy management.

**Integration with IoT**: The integration of RL with IoT devices can enhance real-time data acquisition and decision-making processes in greenhouses. Future research should focus on developing seamless IoT-RL integration frameworks.**Scalability**: Research on scaling RL solutions to larger, more complex greenhouse systems is necessary to ensure widespread adoption. Studies should address computational challenges and the ability to handle large datasets.**Interdisciplinary Approaches**: Combining RL with other AI techniques, such as genetic algorithms and fuzzy logic, could yield more robust energy management solutions. The exploration of hybrid models that leverage the strengths of different AI paradigms is essential.**Environmental Adaptability**: Developing RL algorithms capable of adapting to diverse environmental conditions will be crucial for global applications. This includes designing algorithms that can learn and adapt to changing weather patterns, pest infestations, and other environmental variables.**Economic Viability**: Studies on the cost-effectiveness of RL implementations in greenhouses can drive commercial interest and investment. Future research should focus on performing cost–benefit analyses and developing business models that highlight the economic advantages of RL-based energy management systems.**User-Friendly Interfaces**: Developing user-friendly interfaces and control systems for greenhouse operators is vital for the practical implementation of RL. Research should focus on creating intuitive dashboards and control panels that allow operators to easily interact with and oversee RL systems.**Sustainability Metrics**: Future work should also explore the development of sustainability metrics that RL systems can optimize. This includes not only energy efficiency but also water usage, pesticide application, and overall environmental impact.**Policy and Regulatory Compliance**: Research should address how RL systems can be designed to comply with local and international policies and regulations concerning energy usage and environmental protection.**Data Privacy and Security**: With the increasing use of IoT and RL, ensuring data privacy and security is essential. Future research should develop robust security protocols to protect sensitive data in greenhouse management systems.**Real-World Case Studies**: Conducting real-world case studies and pilot projects can provide valuable insights into the practical challenges and benefits of implementing RL in greenhouses. These studies can help refine RL models and identify best practices for successful adoption.

### 2.2. Literature Review Conclusions

The RL paradigm has shown significant potential to optimize energy management in greenhouses and precision agriculture techniques. From optimal autonomous control to deep RL, the reviewed references indicate a growing trend toward intelligent, adaptive solutions that promote energy efficiency and sustainability. Future research should focus on integrating RL with IoT, scalability, interdisciplinary approaches, environmental adaptability, economic viability, user-friendly interfaces, sustainability metrics, policy compliance, data privacy and security, and real-world case studies to further advance this technological evolution.

In conclusion, the field of reinforcement learning for greenhouse management has made significant progress, yet challenges remain. Future research should focus on improving scalability, integrating advanced technologies, and developing hybrid models to fully realize the potential of RL in this domain.

## 3. Materials and Methods

### 3.1. Model Proposed

The reinforcement learning (RL) problem involves a digital agent exploring an environment to achieve a specific goal. In the field of automated greenhouses, it is about managing environmental conditions and crop growth while optimizing resources and inputs. RL is based on the hypothesis that all goals can be characterized by maximizing the expected cumulative reward. The agent must learn to perceive and manipulate the state of the environment through its actions to optimize this reward. In the model presented in this work, the agent performs control actions to adjust the set-points, ensuring that the internal conditions of the greenhouse stay within the predefined maximum and minimum limits. By doing so, the agent optimizes the management of resources such as water, energy, and inputs, leading to improved plant growth. Figure 1 shows the application scenario of the RL paradigm.

The system must be able to collect relevant data, establish the necessary sensors and communication protocols, and manage them appropriately using the IOT paradigm.

The proposed model uses a layered architecture (Figure 2) to integrate the services and functionalities of the platform (IoT + RL). In an agricultural facility, whether newly constructed or already operational, monitoring and actuation devices are installed and interfaced with the processing and control layer. The following provides a detailed description of each layer.

In the physical layer, IoT sensors/actuators, human–machine interfaces (HMIs) and machine-to-machine interfaces (MMIs) are installed. The farmer and technicians can act by entering data and requests. The processing layer filters data, executes control actions, and communicates with the upper layer.

The control layer embedded devices receive data from the upper layer to manage the different processes and perform control and maintenance actions. This layer must provide the necessary support to communicate the data to the sensor/actuators installed.

The interoperability layer integrates different RL agents connected to digital twin algorithms. The model proposes the development of RL agents to optimize the use of set-point values in various control loops within an automated facility. The RL agent modifies the set-point values to achieve one or more specified objectives, such as maintaining environmental conditions, improving electrical consumption, reducing water use, and optimizing renewable energy use. This process operates automatically and can be analyzed through an interface where users can test control strategies. This interface enables simulations on a digital twin model that replicates different states based on the knowledge acquired from the analyzed data obtained from IoT network sensors. In this layer, the deployment of RL agents, the development of digital twin functionalities, and the implementation of user interfaces are executed.

In the application layer, applications are designed and developed on different platforms (mobile phones, business networks, computers, etc.). In each of them, a relationship is defined between the user and the type of access allowed.

#### 3.1.1. Development in an Automated Greenhouse

Greenhouses provide a controlled environment for plant cultivation, allowing for improved growth and productivity compared to traditional open-field agriculture. However, managing a greenhouse efficiently involves complex decision-making processes to balance resource usage, such as water and energy, with the optimal growing conditions for plants. Traditional management methods often rely on predefined schedules and heuristics, which may not adapt well to dynamic environmental conditions and changing plant needs.

Reinforcement learning (RL) offers a promising solution to this challenge by enabling systems to learn optimal strategies through interactions with the environment. RL algorithms can adapt to changing conditions and learn from experience, making them well suited to the dynamic and complex nature of greenhouse management. By continuously adjusting actions based on observed outcomes, RL can optimize resource usage while maintaining or improving crop yields.

One of the scenarios where the model can be applied is in energy management. In the case of the use of this work, the model is based on a reinforcement learning algorithm in the management of greenhouse energy. Specifically, the model employs a prediction model to forecast the greenhouse’s environmental conditions throughout the day. Based on these predictions, they implement a reinforcement learning (RL) algorithm that rewards minimal energy usage to regulate the greenhouse’s temperature. The algorithm refines the temperature control loop by optimizing the choice of the input point for the existing regulation loop. In essence, the algorithm adjusts the set point value to optimize energy consumption while maintaining the selected minimum and maximum temperature thresholds.

Incorporating climate predictions into the RL model can further enhance its performance by allowing anticipatory adjustments.

Reinforcement learning (RL) offers a promising approach to energy consumption by learning from the environment and making data-driven decisions. This study focuses on the Q-learning algorithm, a model-free RL method, to determine the connection and disconnection of the air conditioning system.

#### 3.1.2. Methodology

The proposed system uses the Q-learning algorithm to make watering decisions based on real-time data from IoT sensors. The states, variables, actions, and reward function are designed to reflect the dynamics of the greenhouse environment. The methods are indicated in Table 1.

### 3.2. Phase 1: Control Strategies (Set-Point in the Environmental Regulation Loop)

Reinforcement learning (RL) techniques can optimize air conditioning connection strategies by making decisions based on continuous feedback from plants and the environment. Various sensors (energy consumption, temperature, humidity, weather data, etc.), actuators, and a data processing unit must be used. In Table 2, different strategies and their relationship with IoT technologies are shown.

The selection of the appropriate irrigation strategy for greenhouse cultivation depends on various factors, including crop type, environmental conditions, and resource availability. Strategies based on substrate moisture, evapotranspiration, and VPD offer customized approaches to optimize water use, improve crop performance, and promote sustainable agricultural practices.

### 3.3. Phase 2: IoT Infrastructure

The IoT infrastructure is designed with the strategy of being interoperable with existing greenhouse subsystems: climate, lighting, irrigation, etc. To achieve this, it is designed and developed at two levels.

Figure 3 describes the Internet of Things (IoT) network. Table 3 contains the identification of the components according to this figure. The sequence of events and data sequence are listed below.

Climate energy data are captured and stored in the system.Sensors collect environmental conditions.The data are sent to the gateway, which then transmits it to the local server and cloud services.The data are stored on the local server. The datasets are created with the main variables.Local and cloud services allow remote monitoring and control of the system.The prediction of the environmental conditions in the next few hours is obtained.With the data and predictions, the learning algorithm is executed.The algorithm proposes a modification of the set-point by increasing or decreasing its value, using the values of the reinforcements or Q values.

**Table 3 sensors-24-08109-t003:** Identification of components in the IoT network for a solar energy management system.

Number	Component Description
1	**IoT Gateway or Router**: This device acts as a central communication point between various sensors and devices and the cloud server. It uses IoT protocols to transmit data.
2	**Sensors and Meters**: These devices collect data from different sources:
2a	Energy meter (A-Wh) that measures the amount of energy consumed.
2b	Temperature sensor (thermostat) that measures the ambient temperature.
3	**Solar Energy Controller**: This component receives data from the sensors and manages the distribution of energy.
3a	Inverter that converts solar energy from direct current (DC) to alternating current (AC).
3b	Batteries for energy storage.
3c	Switches and fuses for protection.
4	**Local Server or Database**: Stores and processes data locally. It is where all the data are collected for processing before being sent to the cloud.
5	**Cloud Services**: The data are sent to the cloud for additional storage and analysis. Cloud services can provide interfaces to monitor and control the system.
M	**Monitoring Computer**: Allows users to interact with the system, probably through a graphical interface for real-time monitoring and control of the system.

### 3.4. Phase 3: Dataset Generation

Datasets generated from IoT sensors provide a rich source of information for decision makers. These datasets enable real-time monitoring and analysis, allowing for informed decision making based on accurate and up-to-date information. In greenhouse facilities, IoT sensor traffic data can be used to optimize traffic flow, reduce congestion, and improve public transportation systems. Generating high-quality datasets from IoT sensor data requires addressing issues related to data quality and consistency. Sensors can produce noisy or incomplete data, which can affect the accuracy and reliability of the datasets. Implementing robust data preprocessing and cleaning techniques is essential to ensure the integrity of the datasets.

Temperature and humidity sensors are located inside and outside the greenhouse, monitoring the weather forecast and energy consumption. The methodology for data capture is outlined below:Sensor Placement. Strategically place sensors to capture a representative sample of environmental conditions within the greenhouse. This includes placing sensors at various depths and locations throughout the greenhouse to monitor microclimates.Data Logging and Transmission. Utilize data loggers and wireless networks to ensure continuous data capture and transmission. This includes setting up a reliable network infrastructure that can handle the data volume and frequency required for RL applications.Data Storage. Implement a centralized data storage solution, preferably cloud-based, to store the large volumes of data generated by the sensors. Ensure the storage system supports efficient data retrieval and processing.Data Prepossessing, Cleaning, and Normalization. Address issues related to missing values, sensor malfunctions, and noise in the data. Techniques such as interpolation and filtering should be applied to ensure data quality and consistency. Normalise sensor data to a common scale to facilitate accurate analysis and model training. This step is crucial for integrating diverse data types into a cohesive dataset.

#### Materials and Methods

Data capture was carried out on a hemp crop in a technical greenhouse (Figure 4) with the following characteristics: glass greenhouse of 50 m^2^ (5 × 10 m) each. All of them have an automatic climate and irrigation control system with the following equipment:Humidifier with osmosis water mist.Air conditioners to heat and cool the modules. Twin, Triple Mitsubishi PUHZ-P200YKA three-phase classic inverter Nominal cooling capacity (Min.–Max.) kW 19.00 Nominal heat capacity (Min.–Max.) kW 22.40.Thermal shading screen.Extractor fan and zenithal opening windows with anti-trip mesh.Artificial light lamps to increase net assimilation.Micro-sprinkler and drip or flood irrigation system.Temperature and humidity probes.Electrical, compressed air, mains water and osmosed water connections.Embedded device (raspberrypi4) that deploys an intranet (WiFi, Bluetooth Low Energy) for communication, monitoring, and control.Electric energy meter in three-phase and single-phase circuits (shelly 3 EM). This consumption meter communicates with the embedded system through the WiFi and IP protocol.Communication to web servers to obtain the temperature prediction in the greenhouse area.

**Figure 4 sensors-24-08109-f004:**
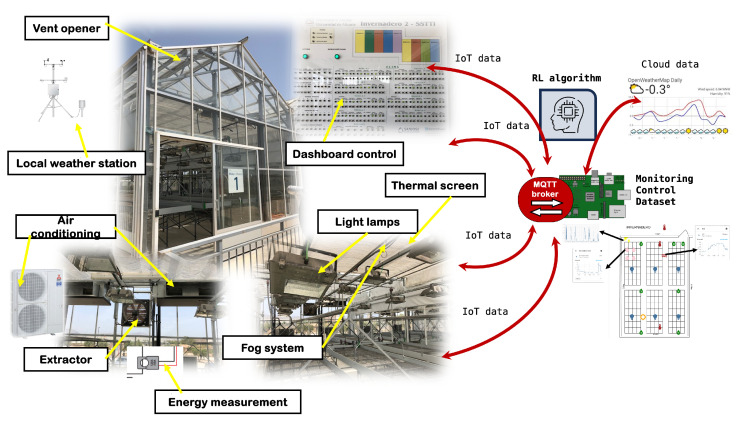
Industrial greenhouse used to deploy the IoT data. A control intranet has been deployed with embedded devices and communications based on IoT protocols (WiFi and MQTT).

This work develops and tests a low-cost sensor/actuator network platform, based on the Internet of Things, integrating machine-to-machine and human–machine interface protocols used in [22]. The system integrated Internet of Things (IoT) paradigms, ubiquitous sensor networks, and edge computing to create a smart agricultural environment. Implementation (Figure 4) included machine-to-machine and human–machine interface protocols to enable seamless communication, data capture (Figure 5) and control processes considering a precision agriculture scenario.

### 3.5. Phase 4: Digital Model and RL Algorithm

RL involves an agent learning to make sequential decisions through interaction with its environment to maximize cumulative rewards. In the context of greenhouse control, RL offers a promising framework to optimize energy use while maintaining the configured maximum and minimum temperature conditions. The theoretical model serves to support decision making at a given moment. The objective is to start from prior knowledge that allows the system to initiate decision making more effectively. From the initial configuration of the RL algorithm, the system will adapt to the specific conditions of the installation, optimizing control as feedback information is obtained on the evolution of the system compared to its previous behavior. It is therefore about introducing a finer regulation that is capable of improving the existing one by adapting to the specific conditions of the installation.

#### 3.5.1. Greenhouse Model Based in Differential Equations

There is knowledge and mathematical models of the behavior of temperature in a greenhouse. These models provide a good starting point for defining reward policies and functions. For all these reasons, the RL model is based on the knowledge already acquired to, from there, introduce improvements and optimize current operation. The behavior of the temperature inside a greenhouse is influenced by several factors, including the external temperature, solar radiation, and the thermal dynamics of the greenhouse itself. Here, we present a mathematical model to simulate the dynamics of the greenhouse temperature.

The thermal model of a greenhouse when both heating and cooling are applied can be described by the following differential equation.
(1)$$C_{\mathrm{greenhouse}}\frac{dT_{\mathrm{in}}\left(t\right)}{dt}=Q_{\mathrm{solar}}\left(t\right)+Q_{\mathrm{heat}}\left(t\right)-Q_{\mathrm{cool}}\left(t\right)-UA\left(T_{\mathrm{in}}\left(t\right)-T_{\mathrm{out}}\left(t\right)\right)$$

To discretize this equation for algorithmic application, we consider a time step Δt. The temperature change over the time step from *t* to t+Δt can be approximated as
(2)$$\frac{dT_{\mathrm{in}}\left(t\right)}{dt}\approx \frac{T_{\mathrm{in}}\left(t+\Delta t\right)-T_{\mathrm{in}}\left(t\right)}{\Delta t}$$

Substituting this into the differential equation gives
(3)$$C_{\mathrm{greenhouse}}\frac{T_{\mathrm{in}}\left(t+\Delta t\right)-T_{\mathrm{in}}\left(t\right)}{\Delta t}=Q_{\mathrm{solar}}\left(t\right)+Q_{\mathrm{heat}}\left(t\right)-Q_{\mathrm{cool}}\left(t\right)-UA\left(T_{\mathrm{in}}\left(t\right)-T_{\mathrm{out}}\left(t\right)\right)$$

Solving for $T_{\mathrm{in}}\left(t+\Delta t\right)$, we obtain the following
(4)$$T_{\mathrm{in}}\left(t+\Delta t\right)=T_{\mathrm{in}}\left(t\right)+\frac{\Delta t}{C_{\mathrm{greenhouse}}}\left[Q_{\mathrm{solar}}\left(t\right)+Q_{\mathrm{heat}}\left(t\right)-Q_{\mathrm{cool}}\left(t\right)-UA\left(T_{\mathrm{in}}\left(t\right)-T_{\mathrm{out}}\left(t\right)\right)\right]$$

The meaning of all variables is shown in Table 4.

This discretized equation can be used to iteratively compute the inside temperature of the greenhouse in discrete time steps for the purpose of applying control algorithms. This model is simple, and the results give us a first analysis of trends and future strategies. In the scenario of an RL algorithm, the actions taken depend on the value of the set temperature. The actions that can be taken are shown in the Table 5.

#### 3.5.2. Greenhouse Model Based in Predictions

This model has two parts. In the first case, the behavior of the greenhouse temperature in the next moments of time is predicted on the basis of IoT data and the machine learning paradigm. Once the future value of the temperature inside the greenhouse is calculated, the best possible action for regulation is taken.

The interior temperature of a greenhouse based on external climatic conditions and historically can be obtained with several models, including linear regression, decision tree, gradient boost, random forest, and neural networks, which can be compared to identify the most accurate and consistent predictive model.

The dataset used consists of climatic data collected from a greenhouse over several months. The variables include the following:**TE**: Exterior Temperature;**HRE**: Exterior Relative Humidity;**RGE**: Exterior Global Radiation;**VV**: Wind Speed;**DV**: Wind Direction;**LL**: Rainfall;**TI**: Interior Temperature;**HRI**: Interior Relative Humidity.

The data are resampled to intervals that are configured according to the installation, and lag features are created to incorporate historical data into the prediction model.

The actions taken are shown in Table 6.

#### 3.5.3. Reinforcement Learning Deployment

Reinforcement learning (RL) is a subfield of machine learning where an agent learns to make decisions by interacting with an environment to maximize a cumulative reward. In the context of controlling the temperature of a greenhouse, an RL agent can learn when to turn the heater on or off to maintain the desired temperature and minimize energy consumption.

The RL model consists of the following elements:**State (*s*)**: Represents the current situation of the environment. In our case, the state can include the internal temperature of the greenhouse, the external temperature, and solar radiation.**Action (*a*)**: The decision taken by the agent in each state. In our case, actions are turning the heater on (a=1) or off (a=0).**Reward (*r*)**: The feedback received by the agent after taking an action in a state. The reward can be a function of the internal temperature and energy consumption.**Policy (π)**: The strategy followed by the agent to make decisions. The policy maps states to actions.**Value (V(s))**: The expected value of the cumulative reward starting from state *s* following policy π.**Q-Value (Q(s,a))**: The expected value of the cumulative reward starting from state *s* taking action *a* and following policy π.

The goal of the RL agent is to learn an optimal policy $\pi^{*}$ that maximizes the cumulative reward. This is formalized in the optimal control problem in a Markov Decision Process (MDP).

##### RL Model Formulas

**State Value Function**:(5)$$V^{\pi }\left(s\right)=E\left[\sum_{t=0}^{\infty }\gamma^{t}r_{t}|s_{0}=s\right]$$
where γ∈[0,1) is the discount factor that weights the importance of future rewards.

**Action Value Function (Q-Value)**:(6)$$Q^{\pi }\left(s,a\right)=E\left[\sum_{t=0}^{\infty }\gamma^{t}r_{t}|s_{0}=s,a_{0}=a\right]$$

**Q-Value Update (Q-Learning)**:(7)$$Q\left(s_{t},a_{t}\right)\leftarrow Q\left(s_{t},a_{t}\right)+\alpha \left[r_{t+1}+\gamma \max_{a}Q\left(s_{t+1},a\right)-Q\left(s_{t},a_{t}\right)\right]$$
where α is the learning rate.

The meaning of all variables is shown in the Table 7.

### 3.6. Phase 5: Training and Methods

To start with prior knowledge and avoid the initial errors introduced by RL, actions, functions, and reward policies are analyzed first through simulation and then using real data obtained in the cultivation process. Firstly, an analysis is carried out using differential equations that model the climatic behavior in a greenhouse and then apply it with the data obtained in the greenhouse.

The method used for the RL algorithm applied to a greenhouse model is based on differential equations.

The RL agent intervenes in the control process through the following steps:**Observe Current State**: The agent observes the current state $s_{t}$, which includes the internal temperature ($T_{in}$), the external temperature ($T_{out}$), and solar radiation ($Q_{solar}$).**Select Action**: Based on its policy π, the agent chooses an action $a_{t}$ (turn the heater on or off).**Apply Action**: The chosen action is applied to the environment.**Observe Reward and Next State**: The agent receives a reward $r_{t}$ and observes the next state $s_{t+1}$.**Update Policy**: The agent updates its policy using the learning algorithm, such as Q-learning.

By formulating temperature control as a sequential decision-making problem, RL algorithms can adaptively adjust based on real-time environmental data and set-point conditions. One crucial aspect in applying RL to temperature control is the choice of state representation, which captures relevant information about the environment for decision making.

The process is based on the technician’s previous configuration and a theoretical model of temperature behavior on which the RL algorithm performs the calculations. This model can be based on energy balances or be a model obtained by studying the behavior of greenhouse conditions.

Selecting an appropriate state representation is essential to allow the RL agent to effectively learn and adapt its cotrol policies to achieve optimal performance under varying environmental conditions. Furthermore, the careful selection of control actions is equally important, as it determines how the RL agent interacts with the environment. By choosing suitable control actions, such as adjusting the temperature set-point, the RL agent can effectively optimize energy usage. Finally, algorithms are developed to optimize the reward functions associated with states and actions, ensuring that the RL agent learns to make decisions that lead to the most favorable outcomes. This approach allows the development of intelligent control systems capable of dynamically responding to changing environmental conditions and ambient conditions needs.

The Q-learning algorithm has five actions (Table 5).

Table 8 describes examples of potential reward functions applicable in RL algorithms.

An appropriate RL algorithm, such as Q-learning, deep Q network (DQN), or posterior policy optimization (PPO), should be chosen based on the complexity of the task and the available computational resources, where the following apply:α, β, and γ are weight coefficients that can be tuned.*E* is the energy consumption penalty.$T_{\mathrm{in}}$ is the current inside temperature.$T_{\mathrm{set}\text{-}\mathrm{point}}$ is the set-point temperature.

D1,D2 is the penalty for temperature deviation.


$$D1=|T_{\mathrm{in}}-T_{\mathrm{set}\text{-}\mathrm{point}}|$$



$$D2=|T_{\mathrm{in}}-T_{\mathrm{set}\text{-}\mathrm{point}}|\;+\;{\left(T_{\mathrm{in}}-T_{\mathrm{set}\text{-}\mathrm{point}}\right)}^{2}$$


L1,L2 are the limit penalty for exceeding the maximum or minimum temperature. The limit penalty is applied when the inside temperature exceeds the maximum or minimum temperature limits:

$$L1=\left\{\begin{array}{cc} |T_{\mathrm{in}}-T_{\max}| & \mathrm{if}\;T_{\mathrm{in}}>T_{\max}\\ |T_{\mathrm{in}}-T_{\min}| & \mathrm{if}\;T_{\mathrm{in}}<T_{\min}\\ 0 & \mathrm{if}\;T_{\min}\leq T_{\mathrm{in}}\leq T_{\max} \end{array}\right.$$$$L2=\left\{\begin{array}{cc} {\left(T_{\mathrm{in}}-T_{\max}\right)}^{2} & \mathrm{if}\;T_{\mathrm{in}}>T_{\max}\\ {\left(T_{\min}-T_{\mathrm{in}}\right)}^{2} & \mathrm{if}\;T_{\mathrm{in}}<T_{\min}\\ 0 & \mathrm{otherwise} \end{array}\right.$$
where $T_{\max}$ and $T_{\min}$ are the maximum and minimum allowable temperatures, respectively.

*S* is the set-point change penalty.The set-point change penalty discourages frequent and large adjustments to the set-point temperature:
$$S=\;|T_{\mathrm{set}\text{-}\mathrm{point}}-T_{\mathrm{previous}\text{\_}\mathrm{set}\text{-}\mathrm{point}}|$$
where $T_{\mathrm{previous}\text{\_}\mathrm{set}\text{-}\mathrm{point}}$ is the set-point temperature from the previous time step.

## 4. Results and Discussion

### 4.1. Results of the Reward Functions for Greenhouse Control

The reward function aims to balance the trade-offs between energy efficiency, temperature stability, and smooth set-point adjustments. By carefully tuning the weights α, β, γ, and δ, the reinforcement learning agent can learn an optimal strategy to control the greenhouse temperature. Three reward functions are indicated in Table 8. In this section, we analyze the result obtained by applying each of them to the temperature evolution model in a greenhouse (Figure 6). From the result obtained, the first conclusions will be drawn to obtain the best initial configuration of the RL algorithm that will act in the installation under real conditions. In this work, we compare the three strategies in Table 5. We also compare the control with RL and the results obtained with a classic control without RL. This analysis and comparison will result in the theoretical improvement of the algorithm compared to the current control and the best-reward policy. Once the theoretical analysis of the best reward function applicable to the theoretical model and greenhouse conditions has been carried out, the algorithm is implemented in the facility using real data and actions defined in the previous step. The comparison of energy consumption between the reinforcement learning (RL)-based control and a fixed set-point control at different sampling times showed varied results. The following analysis examines the percentage difference in energy consumption between systems controlled by reinforcement learning (RL) and those with a fixed set-point. The cooling and heating processes of the greenhouse have been analyzed. Each of the analyses provides a figure with six images with different comparisons. The six subplots in each provided figure are evaluated to determine which system is more energy efficient and effective in maintaining the desired temperature range.

#### 4.1.1. (1a) Cooling Temperature Control Analysis Using the Differential Equation Model (Figure 7)

In R1a-R1b, the RL system manages to maintain the inside temperature within the desired range more effectively than the fixed set-point system. Temperature variations with RL control are less pronounced and stay closer to the set-point compared to the fixed set-point system. In R2a-R2b, similar to R1a-R1b, the RL control maintains a more stable temperature with fewer and smaller deviations from the set-point. The fixed set-point system shows greater fluctuations and occasionally falls outside the desired temperature range. In R3a-R3b, the graph reinforces the previous observations, where the RL system exhibits better control over the temperature, keeping it within the desired limits more consistently than the fixed set-point system. The fixed set-point control shows larger temperature changes and less precision in maintaining the set-point. R1a and R3a appear to perform the best with higher and more consistent energy savings at different sampling times. The top left subplot shows a slight edge in overall savings stability.

**Figure 7 sensors-24-08109-f007:**
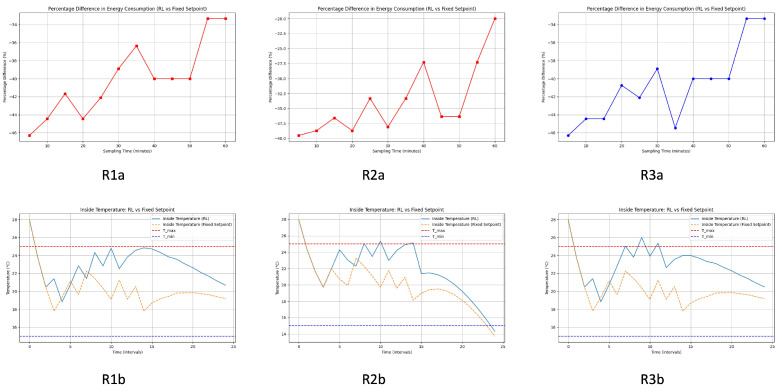
Comparative analysis in greenhouse cooling process: R1a: percentage difference in energy consumption (RL-reward function 1 vs. fixed set-point), R2a: percentage difference in energy consumption (RL-reward function 2 vs. fixed set-point), R3a: percentage difference in energy consumption (RL-reward function 3 vs. fixed set-point), R1b: inside temperature (RL-reward function 1 vs. fixed set-point), R2b: inside temperature (RL-reward function 2 vs. fixed set-point), R3b: inside temperature (RL-reward function 3 vs. fixed set-point).

##### General Improvement with RL Control

Across all energy consumption graphs, the RL system consistently outperforms the fixed set-point system in terms of energy savings. The improvement is particularly notable for shorter sampling times (5 to 20 min), where the savings are more substantial. Even at longer sampling intervals (up to 60 min), the RL system maintains a significant percentage of energy savings.

##### Temperature Control Effectiveness

The lower subplots demonstrate that the RL control system maintains the desired temperature range more effectively than the fixed set-point system. The RL system results in smaller deviations from the set-point, indicating better control and stability. The fixed set-point system shows larger temperature fluctuations and occasionally fails to keep the temperature within the desired range.

#### 4.1.2. Discussion in Cooling Process

The analysis indicates that the RL control system is highly effective in reducing energy consumption compared to the fixed set-point system. Shorter sampling times yield the best results with the RL system showing a 45% improvement in energy efficiency. Furthermore, the RL system demonstrates superior temperature control, maintaining the desired range more consistently and with fewer deviations. In general, the RL system is a preferable choice for optimizing energy use and temperature stability in greenhouse temperature control scenarios.

The R1a-R1b subgraph shows a consistent improvement in energy consumption with the RL system as the sampling time increases. For sampling times between 5 and 10 min, the RL system shows significant energy savings of around −45%. As the sampling time increases, the savings decrease slightly but remain significant, ending around −35% at 60 min. R2a-R2b exhibits a similar trend with the RL system performing better than the fixed set-point system. The initial savings are around −40% with fluctuations as the sampling time increases. The savings dip slightly around 30 to 40 min but show an overall improvement, ending around −20% for 60 min. In R3a-R3b, the trend is also consistent with the other graphs, showing that the RL system consistently saves energy compared to the fixed set-point system. The savings are initially around −45% for shorter sampling times. Despite fluctuations, the savings remain significant throughout, ending around −35% for 60 min.

#### 4.1.3. (1b) Heating Temperature Control Analysis Using the Differential Equation Model (Figure 8)

##### General Improvement with RL Control

Across all energy consumption graphs, the RL system consistently outperforms the fixed set-point system in terms of energy savings. The improvement is particularly notable for shorter sampling times (5 to 20 min), where the savings are more substantial. Even at longer sampling intervals (up to 60 min), the RL system maintains a significant percentage of energy savings.

**Figure 8 sensors-24-08109-f008:**
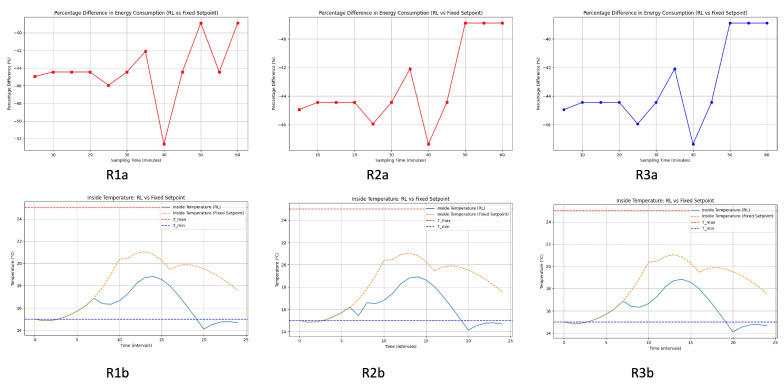
Comparative analysis in greenhouse heating process. R1a: percentage difference in energy consumption (RL-reward function 1 vs. fixed set-point), R2a: percentage difference in energy consumption (RL-reward function 2 vs. fixed set-point), R3a: percentage difference in energy consumption (RL-reward function 3 vs. fixed set-point), R1b: inside temperature (RL-reward function 1 vs. fixed set-point), R2b: inside temperature (RL-reward function 2 vs. fixed set-point), R3b: inside temperature (RL-reward function 3 vs. fixed set-point).

##### Temperature Control Effectiveness

The bottom subplots demonstrate that the RL control system maintains the desired temperature range more effectively than the fixed set-point system. The RL system results in smaller deviations from the set-point, indicating better control and stability. The fixed set-point system shows larger temperature fluctuations and occasionally fails to keep the temperature within the desired range.

#### 4.1.4. Discussion in Heating Process

The analysis indicates that the RL control system is highly effective in reducing energy consumption compared to the fixed set-point system. Shorter sampling times yield the best results with the RL system showing a 45% improvement in energy efficiency. Furthermore, the RL system demonstrates superior temperature control, maintaining the desired range more consistently and with fewer deviations. In general, the RL system is a preferable choice for optimizing energy use and temperature stability in greenhouse temperature control scenarios.

### 4.2. Results for Greenhouse Control Based in Temperature Prediction and Reinforcement Learning

The first goal is to design and implement a model that predicts indoor temperature (*TI*) based on various environmental variables. The model takes advantage of past data and predicted future values to make accurate forecasts. Once there is an estimated value of the temperature evolution inside the greenhouse, the RL algorithm is applied acting on the set-point, depending on said evolution. From the previous study using the differential equations model, it was concluded that the reward function R1 is viable to display the algorithm. For all these reasons, in this analysis with data obtained in the greenhouse, we are going to use this reward function.

#### 4.2.1. Prediction Model

The model predicts the indoor temperature (*TI*) using the following variables:External Temperature (*TE*);External Relative Humidity (*HRE*);Wind Direction (*DV*);Wind Speed (*VV*);External Global Radiation (*RGE*);Internal Relative Humidity (*HRI*).

The prediction is based on the past values (lags) of these variables for the previous 60 min and the predicted external temperatures for the next 60 min.

We created lagged characteristics for the last 60 min and leading characteristics for the future external temperature (*TE*). This allows the model to capture temporal dependencies. The dataset is divided into training and testing sets. The features are standardized to have a mean of 0 and a standard deviation of 1. We use a linear regression model to predict the indoor temperature.

For predicting the interior temperature of the greenhouse, random forest and linear regression models are the most reliable based on their lower RMSE and MAE values. The neural network model’s high error rates suggest that it may not be suitable for this particular prediction task without further tuning or perhaps a different architecture or feature set. Linear regression is chosen for its simplicity and interpretability. The model’s performance is evaluated using the root mean squared error (RMSE) and mean absolute error (MAE) on both the training and testing datasets.

#### 4.2.2. Convergence of the Reinforcement Learning (RL) Method

Figure 7 (cooling process) and Figure 8 (heating process) illustrate that the RL-based control consistently outperforms fixed set-point control in maintaining the desired temperature range while optimizing energy consumption The graphs demonstrate that the RL system stabilizes over time, as shown by the reduced temperature and energy consumption fluctuations after initial adjustments. This stabilization indicates the RL algorithm’s convergence in heating and cooling scenarios. Although the figures do not explicitly plot cumulative rewards over training episodes, the consistent performance improvements and stability of energy savings across different sampling times indirectly confirm that the RL agent has learned an optimal policy, satisfying convergence criteria.

Figure 9 compares the performance of different machine learning models based on two metrics: root mean squared error (RMSE) and mean absolute error (MAE). Data were captured from November 2023 to May 2024.

#### 4.2.3. (2a) Cooling Control Analysis Using the Temperature Prediction Model (Figure 10)

The RL agent was trained to control the greenhouse temperature. The performance of the RL-based control was compared to that of a fixed set-point control. The total energy consumption and the inside temperature regulation were evaluated.

The actions selected depend on the predicted temperature. In this way, it does not depend on a value set in the set-point but rather on variable values that depend on the behavior of the greenhouse.

**Figure 10 sensors-24-08109-f010:**
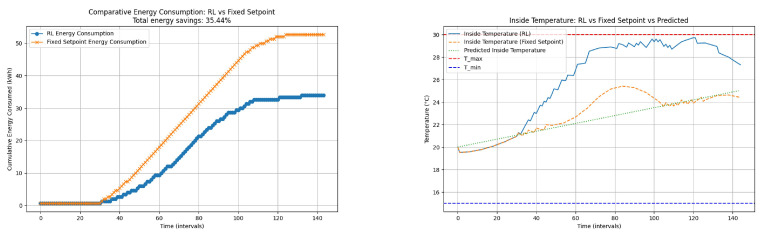
Cooling process: energy consumption: RL vs. fixed set-point (**left**) and inside temperature: RL vs. fixed set-point vs. predicted (**right**). The graph on the left shows comparative energy consumption: RL vs. fixed set-point. The graph on the right shows the inside temperature: RL vs. fixed set-point vs. predicted.

Figure 10 shows the regulation of inside temperature using RL control, fixed set-point control, and predicted inside temperature. The temperature in the RL system is approaching the established maximum more than the temperature set by the set-point. This dynamic adjustment helps minimize energy consumption while ensuring that the temperature stays within the desired range. The predicted temperature provides a reference for the RL control actions. The cumulative energy consumption graph (the graph on the right) demonstrates that the RL-based control performs better than the fixed set-point control, achieving a total energy savings of 35.44%. This significant reduction in energy usage highlights the efficiency of the RL algorithm in optimizing the control strategy. The graph on the right reveals that the RL control effectively maintains the temperature within the desired range (between T_min and T_max). The RL control shows a more dynamic response compared to the fixed set-point control, which can lead to more efficient heating. The predicted temperature aligns well with the actual inside temperature regulated by the RL control, indicating accurate predictions and effective control actions. The RL-based temperature control (in the cooling phase) method shows significant advantages over fixed set-point control in terms of energy savings and effective temperature regulation. The ability of RL to adaptively adjust the set-point temperature based on predicted inside temperatures leads to optimized energy consumption and better maintenance of the desired temperature range.

#### 4.2.4. (2b) Heating Control Using the Temperature Prediction Model (Figure 11)

Figure 11 compares the cumulative energy consumption between the RL-based control and the fixed set-point control. The RL control demonstrates significant energy savings with a total reduction of 25.93% compared to the fixed set-point control. This indicates the effectiveness of the RL algorithm in optimizing energy usage while maintaining the desired temperature.

**Figure 11 sensors-24-08109-f011:**
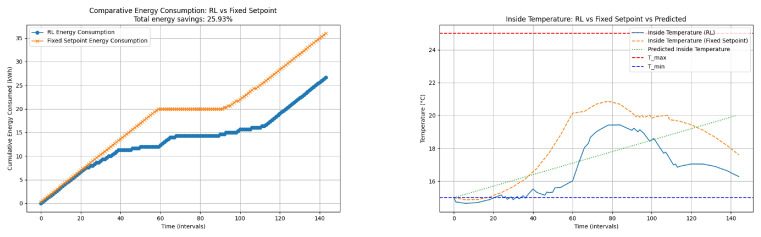
Heating process: comparative energy consumption: RL vs. fixed set-point and inside temperature: RL vs. fixed set-point vs. predicted. The graph on the (**left**) shows comparative energy consumption: RL vs. fixed set-point. The graph on the (**rigth**) shows inside temperature: RL vs. fixed set-point vs. predicted.

The second figure in the composite image shows the regulation of the inside temperature using RL control, fixed set-point control, and the predicted inside temperature. The RL control maintains the temperature closer to the upper limit ($T_{\max}$) compared to the fixed set-point control. This dynamic adjustment helps minimize energy consumption while ensuring that the temperature stays within the desired range. The predicted temperature provides a reference for the RL control actions.

The cumulative energy consumption graph demonstrates that the RL-based control performs better than the fixed set-point control, achieving a total energy savings of 25.93%. This significant reduction in energy usage highlights the efficiency of the RL algorithm in optimizing the control strategy.

The inside temperature regulation graph reveals that the RL control effectively maintains the temperature within the desired range (between $T_{\min}$ and $T_{\max}$). The RL control shows a more dynamic response compared to the fixed set-point control, which can lead to more efficient heating. The predicted temperature aligns well with the actual inside temperature regulated by the RL control, indicating accurate predictions and effective control actions.

The RL-based temperature control method shows significant advantages over the fixed set-point control in terms of energy savings and effective temperature regulation. The ability of RL to adaptively adjust the set-point temperature based on predicted inside temperatures leads to optimized energy consumption and better maintenance of the desired temperature range. The actions taken are shown in Table 9

This Q-learning algorithm optimizes (in the heating phase) the greenhouse set-point temperature by selecting from five actions to adjust the set-point. The reward function penalizes heating usage, encouraging the algorithm to find a set-point that minimizes energy consumption while maintaining the desired temperature range.

#### 4.2.5. Discussion of Energy Consumption and Savings for Different $T_{set-point}$

If the set-point is modified, there may be a specific moment in which the savings could be similar (Figure 12, on the left). However, regulation with a fixed set-point has two difficulties:
If there is a change in the external climate conditions, the temperature is not regulated correctly. The fixed set-point system must be modified in real time.The regulation with a fixed set-point depends on the knowledge of the technician and does not consider external environmental values.

**Figure 12 sensors-24-08109-f012:**
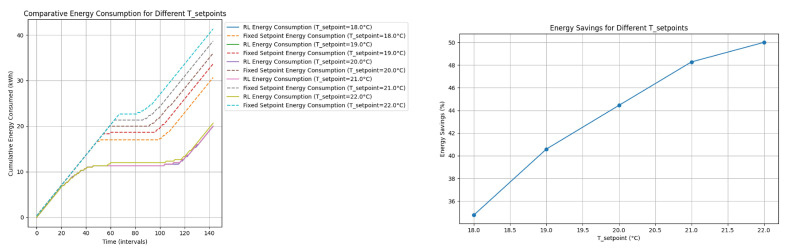
Comparative analysis taking different reference temperatures (set-points). The graph on the (**left**) shows comparative energy consumption for different T_setpoints. The graph on the (**right**) shows energy savings for different T_setpoints.

RL regulation creates a model in which the system is automatically regulated also considering external conditions and installation capacity.

The graph on the left of Figure 12 illustrates the cumulative energy consumption over time for different set-point temperatures ($T_{set-point}$). The graph compares the energy consumption between reinforcement learning (RL)-based control and a fixed set-point control for various set-points.

##### Key Observations


**Energy Consumption Patterns:**
-The energy consumption increases over time for both RL and fixed set-point controls.-For lower $T_{set-point}$ values, the energy consumption is generally lower. As $T_{set-point}$ increases, energy consumption increases.
**Comparison between RL and Fixed Set-Point:**
-RL control consistently consumes less energy compared to fixed set-point control for all set-points.-The difference in energy consumption between the RL and the fixed set-point control is more pronounced at higher set-points.

##### Right Graph: Energy Savings for Different $T_{set-point}$

The graph on the right of Figure 12 shows the percentage of energy savings achieved by using the RL control compared to the fixed set-point control in different $T_{set-point}$ values.

##### Key Observations


**Energy Savings Trend:**
-The energy savings increase with higher $T_{set-point}$ values.-The energy savings range from approximately 36% at $T_{set-point}=18.0\;{}^{\circ}C$ to nearly 50% at $T_{set-point}=22.0\;{}^{\circ}C$.
**Efficiency of RL Control:**
-The RL control becomes more efficient in terms of energy savings as the $T_{set-point}$ increases.-This indicates that RL control is particularly beneficial in scenarios where higher set-points are required, resulting in significant energy savings.

The analysis of the graphs reveals that the RL-based control offers substantial energy savings compared to the fixed set-point control across various set-points. The energy savings are more pronounced at higher set-points, indicating the effectiveness of RL control in optimizing energy consumption while maintaining desired temperature conditions in the greenhouse. This makes RL control a promising approach for energy-efficient temperature regulation.

## 5. Conclusions

The results obtained in this work align with the findings of previous studies [1] which also highlight the potential of RL to improve energy efficiency in complex systems such as greenhouses. This work provides an expanded method to apply the integration of RL with IoT considering the participation and knowledge of users (technicians and farmers), the use of non-proprietary hardware and communication protocols. The model can also be implemented both in operating facilities and in new designs.

The integration of Internet of Things (IoT) protocols and reinforcement learning (RL) methodologies has been shown to be effective in managing and optimizing greenhouse operations for industrial hemp cultivation. This combination not only enhances operational efficiency but also maintains selected temperatures and optimizes energy consumption more effectively than classical control methods. By reducing the need for constant human intervention, this technological integration minimizes labor costs and increases scalability for larger agricultural enterprises.

The RL-based control system shows significant energy savings while maintaining the desired temperature ranges, outperforming traditional fixed set-point control systems. Specifically, the study shows energy savings of up to 45% during cooling processes and 25.93% during heating processes. In addition, this new control approach simplifies the workload of technicians by eliminating the need for complex analyses to achieve the same results, allowing them to focus on higher-level oversight and maintenance tasks.

Integration with IoT plays a crucial role in this setup, enabling real-time data acquisition and seamless communication between various greenhouse subsystems. IoT devices collect and transmit environmental data that RL algorithms use to make informed dynamic adjustments to greenhouse conditions. This IoT integration ensures that the RL model can adapt to changing conditions promptly and accurately, thus optimizing resource use and improving overall system responsiveness.

RL algorithms are capable of adaptively adjusting set-point temperatures based on real-time data and predictions, leading to optimized energy consumption and better maintenance of desired environmental conditions. The study validates the practical implementation of RL models in automated greenhouses in the real world, showcasing their ability to scale and adapt to different types of crops and environmental conditions. These conclusions highlight the potential of combining IoT and RL technologies to improve the efficiency, scalability, and sustainability of greenhouse operations, particularly for industrial hemp cultivation.

## Figures and Tables

**Figure 1 sensors-24-08109-f001:**
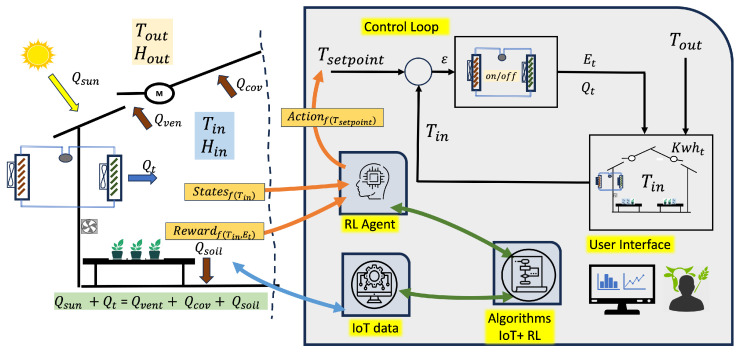
RL paradigm in greenhouse proposed.

**Figure 2 sensors-24-08109-f002:**
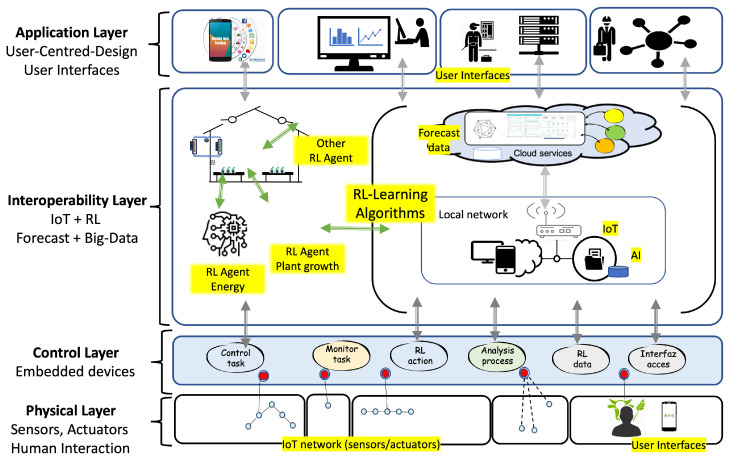
Layered technological architecture. Relationship between IoT, RL, digital platform and different interfaces.

**Figure 3 sensors-24-08109-f003:**
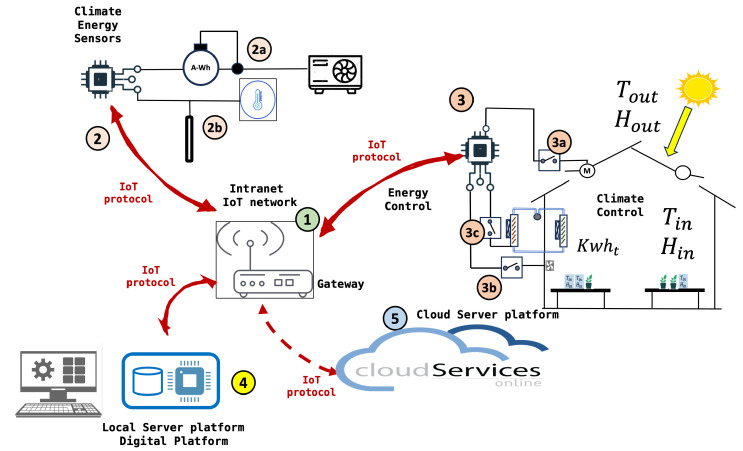
IoT platform. Basic infrastructure for data capture, analysis and management services.

**Figure 5 sensors-24-08109-f005:**
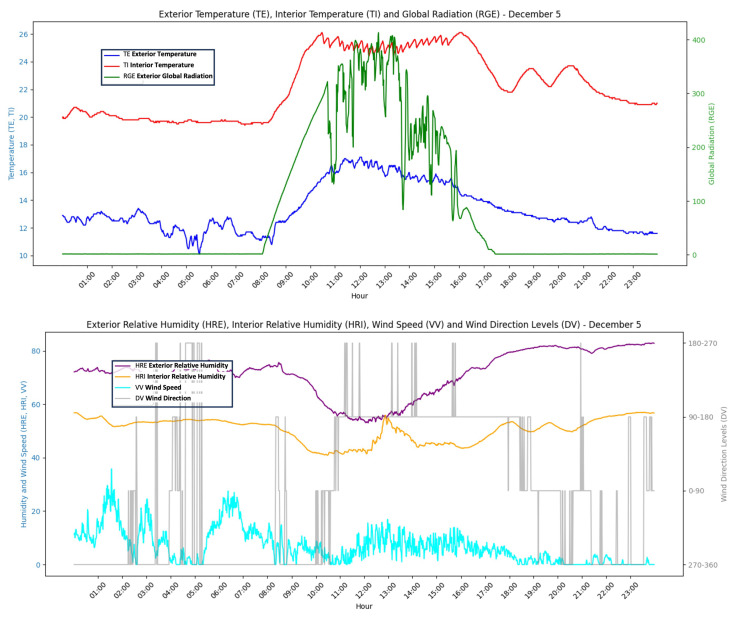
Example of daily data obtained in the dataset. The data are obtained daily every minute and stored in a dataset to obtain temperature prediction models inside the greenhouse. The top graph indicates exterior temperature, interior temperature and global radiation. The figure below indicates exterior relative humidity, interior relative humidity, wind speed and wind direction.

**Figure 6 sensors-24-08109-f006:**
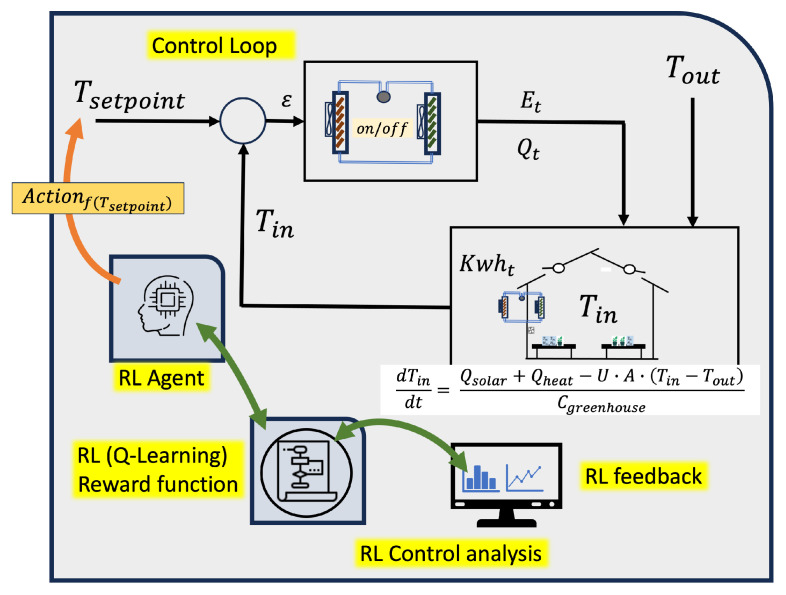
Comparative analysis between RL control, Table 5 reward functions and RL control.

**Figure 9 sensors-24-08109-f009:**
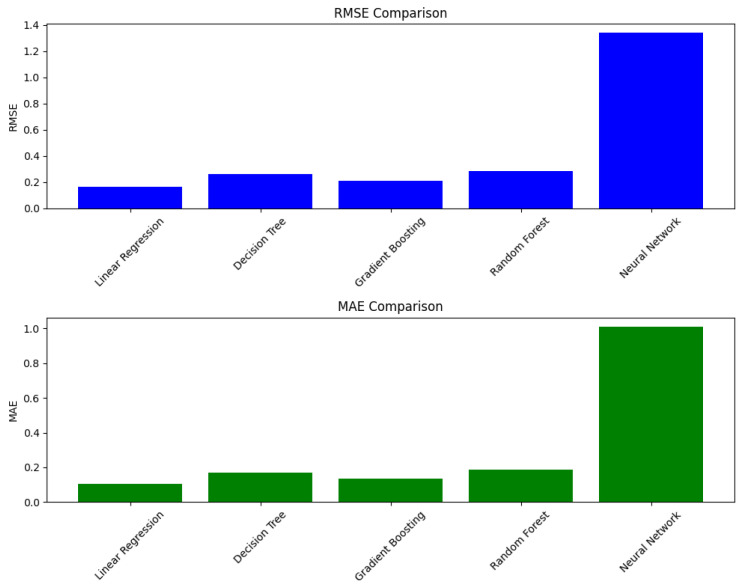
Machine learning methods to predict the temperature inside the greenhouse (TI). The figure above is the RMSE comparison. The figure below is the MAE comparison.

**Table 1 sensors-24-08109-t001:** Method phases.

Phase	Description
Phase 1	**Control strategies**. Analysis of environmental and control variables with the agronomic expert. Examination of all possible strategies that can be part of optimization. Strategies for choosing set-points.
Phase 2	**IoT infrastructure**. Design of the IoT infrastructure needed for the greenhouse to carry out the control and dataset generation. Deployment of various embedded systems interconnected with the required IoT technologies.
Phase 3	**Dataset generation**. Sensors generate data that are analyzed to determine greenhouse behavior models. Each greenhouse has specific characteristics that must be taken into account when applying the model. Datasets are captured in normalized format (csv, json)
Phase 4	**Digital model and an RL algorithm**. The objective is to maintain the greenhouse temperature between the minimum and maximum limits by optimizing the connection and disconnect of the air conditioning system. Implementation by comparing all strategies with those proposed in the model to determine their effectiveness. To act on the installation, a digital model is previously created on which the RL algorithm begins its calculations. This model will be adjusted to the reality of the behavior in the greenhouse application. Theoretical analysis validates reward strategies and policies that will be applied.
Phase 5	**Training and evaluation.** Train the RL agent using the constructed dataset and the results of the analysis, iteratively updating the policy based on observed rewards and state transitions. With the dataset obtained with the capture of IoT data and the results of the theoretical simulations, the policies, reward functions, and actions most appropriate to the type of greenhouse and installation are promoted.

**Table 2 sensors-24-08109-t002:** Energy control strategies. standard control vs. RL control.

Strategy	Description	IoT Sensors
Set-point selection (standard control)	This is the simplest and most commonly used strategy. The technician selects the temperature and the maximum and minimum values	Temperature sensors
Set-point adjustment with RL algorithm (RL actions) to optimize energy consumption	The values assigned at the set-point are adjusted and modified by predicting expected conditions in the greenhouse. These changes are made at scheduled sampling times	Temperature, energy consumption, weather forecast, and temperature prediction inside the greenhouse

**Table 4 sensors-24-08109-t004:** Description of the variables involved in Equations (Equation 1)–(Equation 4).

Variable	Meaning	Units
$C_{\mathrm{greenhouse}}$	Heat capacity of the greenhouse	(J/°C)
$T_{\mathrm{in}}\left(t\right)$	Inside temperature of the greenhouse at time *t*	(°C)
$T_{\mathrm{in}}\left(t+\Delta t\right)$	Inside temperature of the greenhouse at time t+Δt	(°C)
$T_{\mathrm{out}}\left(t\right)$	Outside temperature at time *t*	(°C)
$Q_{\mathrm{solar}}\left(t\right)$	Solar radiation entering the greenhouse at time *t*	(W)
$Q_{\mathrm{heat}}\left(t\right)$	Heating power applied to the greenhouse at time *t*	(W)
$Q_{\mathrm{cool}}\left(t\right)$	Cooling power applied to the greenhouse at time *t*	(W)
*U*	Overall heat transfer coefficient	(W/m^2^ °C)
*A*	Surface area of the greenhouse	(m^2^)
Δt	Time step	(s)

**Table 5 sensors-24-08109-t005:** Actions example taken by RL algorithm using differential equations. Values can be modified depending on the needs of the installation (cooling or heating). The number of actions can also be modified.

Action 1	$T_{set-point}=T_{set-point}+2$
Action 2	$T_{set-point}=T_{set-point}+1$
Action 3	$T_{set-point}=T_{set-point}$
Action 4	$T_{set-point}=T_{set-point}-1$
Action 5	$T_{set-point}=T_{set-point}-2$

**Table 6 sensors-24-08109-t006:** Actions example taken by RL algorithm using temperature $\left(T_{in}\right)$ prediction. The control values are limited by $T_{max}$ and $T_{min}$. Values can be modified depending on the needs of the installation (cooling or heating). The number of actions can also be modified.

Action 1	$T_{set-point}$ = $T_{in\_predicted}+2$
Action 2	$T_{set-point}$ = $T_{in\_predicted}+1$
Action 3	$T_{set-point}$ = $T_{in\_predicted}$
Action 4	$T_{set-point}$ = $T_{in\_predicted}-2$
Action 5	$T_{set-point}$ = $T_{in\_predicted}-1$

**Table 7 sensors-24-08109-t007:** Description of the variables involved in Equations (Equation 5)–(Equation 7).

Variable	Meaning
**State (*s*)**	Represents the current situation of the environment. In our case, the state can include the internal temperature of the greenhouse, the external temperature, and solar radiation
**Action (*a*)**	The decision taken by the agent in each state. In our case, actions are turning the heater on (a=1) or off (a=0)
**Reward (*r*)**	The feedback received by the agent after taking an action in a state. The reward can be a function of the internal temperature and energy consumption
**Policy (π)**	The strategy followed by the agent to make decisions. The policy maps states to actions
**Value (V(s))**	The expected value of the cumulative reward starting from state *s* following policy π
**Q-Value (Q(s,a))**	The expected value of the cumulative reward starting from state *s* taking action *a* and following policy π

**Table 8 sensors-24-08109-t008:** Different reward functions used.

Strategy	Reward Function Proposal
Reward function penalizes the agent for using the heating system	$$R1=\left\{\begin{array}{cc} -1 & \mathrm{if}\;Q_{\mathrm{heating}}>0\\ 0 & \mathrm{if}\;Q_{\mathrm{heating}}=0 \end{array}\right.$$
Reward function penalizes the use of the heating system taking into account the efficiency and the actual deviation from the desired temperature range	R2=−(α·E+β·D1+γ·L1)
Refined reward function: energy consumption penalty. Temperature stability. Exceeding maximum and minimum temperatures. Penalty for frequent changes in the set-point	R3=−(α·E+β·D2+γ·L2+δ·S)

**Table 9 sensors-24-08109-t009:** Actions taken by RL algorithm in heating case using temperature (TI) prediction. The control values are limited by $T_{max}$ and $T_{min}$.

Action 1	$T_{set-point}$ = $T_{in\_predicted}$
Action 2	$T_{set-point}$ = $T_{in\_predicted}-1$
Action 3	$T_{set-point}$ = $T_{in\_predicted}-2$
Action 4	$T_{set-point}$ = $T_{in\_predicted}-3$
Action 5	$T_{set-point}$ = $T_{in\_predicted}-4$

## Data Availability

Data are contained within the article.

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
