# Peer review of "Enhancing Greenhouse Efficiency: Integrating IoT and Reinforcement Learning for Optimized Climate Control"

_sensors, 2024, doi:10.3390/s24248109_

Round 1
Reviewer 1 Report
Comments and Suggestions for Authors
The paper entitled "Enhancing Greenhouse Efficiency: Integrating IoT and Reinforcement Learning for Optimized Climate Control" proposes a novel framework designed to optimize climate control in greenhouse. In this study authors are based on Reinforcement Learning and IoT technology to establish their framework. The obtained results demonstrates that the proposed framework achieves important results. The proposition is important, interesting and attractive. However, authors must do changes.
Modification to do:
1)The introduction is not well written, Author have to rewrite it and describe the issue and some background the importance of the integrating of IoT and Reinforcement Learning in Greenhouse. Furthermore, it should briefly describe how the rest of the paper is organized.
2) the two first paragraphs of section two are duplicated (please delete one)
3) in line 45 LSTM is the abbreviation of "Long Short-Term Memory" not "long-term memory"
4) write "dataset" not "data set" example: in the table 1, section 6
5) why you have only one subsection "6.1. Materials and methods" in the section 6
6) which figure you want to cite in the line 292 (Figure ??)
7) in order to discuss the important of your work, we recommend that you compare your obtained results with the results of related researchers
Author Response
Thanks for the contributions and corrections to the work. They have undoubtedly added quality and clarity to the document
Modification to do:
1)The introduction is not well written, Author have to rewrite it and describe the issue and some background the importance of the integrating of IoT and Reinforcement Learning in Greenhouse.
Corrected in lines 20-32
Furthermore, it should briefly describe how the rest of the paper is organized.
Added in lines 33-44
2) the two first paragraphs of section two are duplicated (please delete one)
Deleted in lines 60-61
3) in line 45 LSTM is the abbreviation of "Long Short-Term Memory" not "long-term memory"
Corrected in line 70
4) write "dataset" not "data set" example: in the table 1, section 6
Corrected in line table 1 in blue color
5) why you have only one subsection "6.1. Materials and methods" in the section 6
Deleted subsection
6) which figure you want to cite in the line 292 (Figure ??)
Corrected in line 317
7) in order to discuss the important of your work, we recommend that you compare your obtained results with the results of related researchers
Added in lines 684-690

Reviewer 2 Report
Comments and Suggestions for Authors
The topic of the paper is quite interesting and the paper reads very well at the beginning, where the studies are nicely motivated and the related literature is analyzed. However, as the paper gets more technical the quality of the presentation and also the content are decreasing more and more.
My particular comments are as follows:
1. In the theoretical parts of the paper, there are a number of equations, which is fine. You should, however, describe the meaning of all variables and the overall meaning of the equations within the text, ideally next to the equations themselves.
2. On page 19 is written that "RL control maintains the temperature closer to the upper limit (Tmax) compared to the fixed setpoint control." However, in the corresponding Fig. it is the other way around.
3. Results like the one in Fig. 11 are not conclusive for deciding whether or not RL performs better than the fixed approach in my opinion. What I mean by that is that if in the fixed setting a lower temp is chosen, less heating leads to less energy consumption, and now why exactly is that worse than using an ML approach to adjust heating? Maybe RL saves energy only since heating is turned up too much in the comparison case?
4. Smaller issues: There are several writing issues, such as the caption of Table 1, the Figure ?? on page 9, the legends in some figs such as Fig. 5 (put text instead of abbreviations as there is plenty of space for it), Font sizes e.g. in Fig. 7, etc. Also the list of Abbreviations before the references is meaningless.
Summarizing, the reviewer would like to encourage the authors to work with more precision and consideration of details, and write the manuscript in a significantly higher quality. In its current shape, with all the issues, the only possible recommendation is a reject.
Author Response
Thanks for the contributions and corrections to the work. They have undoubtedly added quality and clarity to the document
My particular comments are as follows:
1. In the theoretical parts of the paper, there are a number of equations, which is fine. You should, however, describe the meaning of all variables and the overall meaning of the equations within the text, ideally next to the equations themselves.
Thanks for the contribution, all the variables are described in the tables 4 and 7 (red color)
- On page 19 is written that "RL control maintains the temperature closer to the upper limit (Tmax) compared to the fixed setpoint control." However, in the corresponding Fig. it is the other way around.
In the paragraph I want to emphasize that with RL the temperature is closest to Tmax. As you can see in the graph it is correct. I put in the graph what I want to refer to. I don't know if I expressed it wrong.
I changed “The RL control maintains the temperature closer to the upper limit (Tmax) compared to the fixed set-point control” by “The temperature in the RL system is approaching the established maximum more than the temperature set by the setpoint” in lines 591-592
- Results like the one in Fig. 11 are not conclusive for deciding whether or not RL performs better than the fixed approach in my opinion. What I mean by that is that if in the fixed setting a lower temp is chosen, less heating leads to less energy consumption, and now why exactly is that worse than using an ML approach to adjust heating? Maybe RL saves energy only since heating is turned up too much in the comparison case?
Good assessment, this effect is analized in figure 12. To expand the explanation I have added lines 641 to 649
- Smaller issues:
There are several writing issues, such as the caption of Table 1,
the Figure ?? on page 9,
(Corrected)
the legends in some figs such as Fig. 5 (put text instead of abbreviations as there is plenty of space for it),
(abbreviations are shown in lines 363-370. I indicate this reference in te caption of the figure 5.
Font sizes e.g. in Fig. 7.
Also the list of Abbreviations before the references is meaningless.
Added (line 738)
